# Rapid subduction initiation and magmatism in the Western Pacific driven by internal vertical forces

B. Maunder [1✉], J. Prytulak[2], S. Goes [1] & M. Reagan [3]

Plate tectonics requires the formation of plate boundaries. Particularly important is the enigmatic initiation of subduction: the sliding of one plate below the other, and the primary driver of plate tectonics. A continuous, in situ record of subduction initiation was recovered by the International Ocean Discovery Program Expedition 352, which drilled a segment of the fore-arc of the Izu-Bonin-Mariana subduction system, revealing a distinct magmatic progression with a rapid timescale (approximately 1 million years). Here, using numerical models, we demonstrate that these observations cannot be produced by previously proposed horizontal external forcing. Instead a geodynamic evolution that is dominated by internal, vertical forces produces both the temporal and spatial distribution of magmatic products, and progresses to self-sustained subduction. Such a primarily internally driven initiation event is necessarily whole-plate scale and the rock sequence generated (also found along the Tethyan margin) may be considered as a smoking gun for this type of event.

[1] Imperial College London, Department of Earth Science and Engineering, Royal School of Mines, Prince Consort Road, South Kensington, London SW7 2BP, UK. [2] Durham University, Department of Earth Sciences, Science Labs, Lower Mountjoy, South Road, Durham DH1 3LE, UK. [3] University of Iowa, Department of Earth and Envitonmental Sciences, 115 Trowbridge Hall, Iowa City, IA 52240, USA. ✉email: b.maunder@imperial.ac.uk

Plate tectonics sculpts the surface of the Earth and facilitates heat and mass transfer between the exterior and interior of the planet. The formation of continents, the generation of earthquakes and volcanism, and Earth's internal water cycle necessary for planet habitability all fundamentally rely on plate movements. Subduction is the primary driver of plate tectonics[1], and has been a focus of intense research since the inception of plate tectonic theory. However, an understanding of how subduction initiates remains elusive. This is largely because there are few examples of ongoing subduction initiation and none are on a whole-plate scale. Investigations of subduction initiation have largely relied on dynamic numerical models together with the fragmented geologic record (e.g. Stern and Gerya[2]). Here, we take

advantage of the first in situ record of subduction initiation recovered by IODP Expedition 352[3] (Fig. 1a) to construct geologically and geochemically constrained dynamic models.

The most complete magmatic record of subduction initiation has been found at the Izu-Bonin-Mariana subduction zone (IBM)[4]. Dredging and diving in this region[5] suggests that subduction initiation along the full length of the arc started with the production of a magma closely resembling those formed at mid-oceanic ridges, termed "fore-arc basalt" (FAB)[6]. FABs are followed by boninites, which derive from a shallow, depleted mantle[7]. Large fluid fluxes from the subducting plate are required to permit melting of such a depleted source. Boninite transitions towards normal arc tholeiitic to calc-alkaline lava compositions once deep subduction is established[8].

High-precision dating of drill cores from Exp. 352 has revealed that the duration of FAB magmatism, prior to boninitic magmatism is only 0.6–1.2 Myr[9]. The FAB recovered by dredging is of similar age along the full arc, indicating that FAB magmatism occurred along the entire IBM system, near simultaneously (within 1 Myr)[5]. Thus, subduction must have initiated along the IBM in a single, rapid (for such a tectonic process) event. An event of this scale would have far reaching consequences and may have been the catalyst for a global reorganisation of plate motions, observed at this time at the beginning of the Cenozoic[10]. Even if, as alternatively suggested[11], the start of this plate reorganisation triggered IBM initiation, the rapid formation of this new subduction zone likely contributed to the scale of reorganisation.

The explanation proposed for the FAB-boninite sequence is the conceptual model depicted in Fig. 1b–d. We term this process "vertically driven" subduction initiation as it is predominantly driven by vertical buoyancy forces, internal or local to the system, as opposed to "horizontally forced" subduction initiation, driven by horizontal and far-field forces. Vertically driven subduction initiation is often referred to as "spontaneous"[12], however it is not necessarily strictly spontaneous and thus this term can be misleading. For IBM initiation, in the first instance, either a young plate, or a plate that is relatively buoyant due to the presence of a relic arc[13] or rejuvenation by the impact of a mantle plume[14], is juxtaposed with a significantly older, denser, plate by a transform fault with a large offset. At some critical point, perhaps due to a relatively small catalysing vertical force, the old plate sinks into the mantle, creating a lithospheric gap into which hot asthenosphere flows and melts via decompression to form the volatile-poor, mid-ocean-ridge-like FAB. This gap then closes as subduction continues "down-dip" (the sinking plate moving along its length), upon which the subducting plate begins to release fluid, marking the end of FAB formation and the beginning of boninite formation. The Hf–Nd isotopic composition of the first erupted boninites from Exp. 352 also strongly suggest a component of slab melting[15].

The vertically driven conceptual model has been partially tested dynamically[16–19] and it has been shown that the process can occur on a rapid, ~1 Myr, timescale under reasonable conditions[20]. With our new numerical models of IBM subduction initiation, we are now able to demonstrate that this process results in the production of the exact magmatic sequence recovered from the fore-arc and back-arc, both temporally and spatially. This is strong evidence that subduction initiation at the IBM was indeed vertically driven by internal buoyancy. The models also demonstrate that FAB generation does not take place if far-field horizontal compressive forces dominate and is therefore unique to this vertically driven type of subduction initiation.

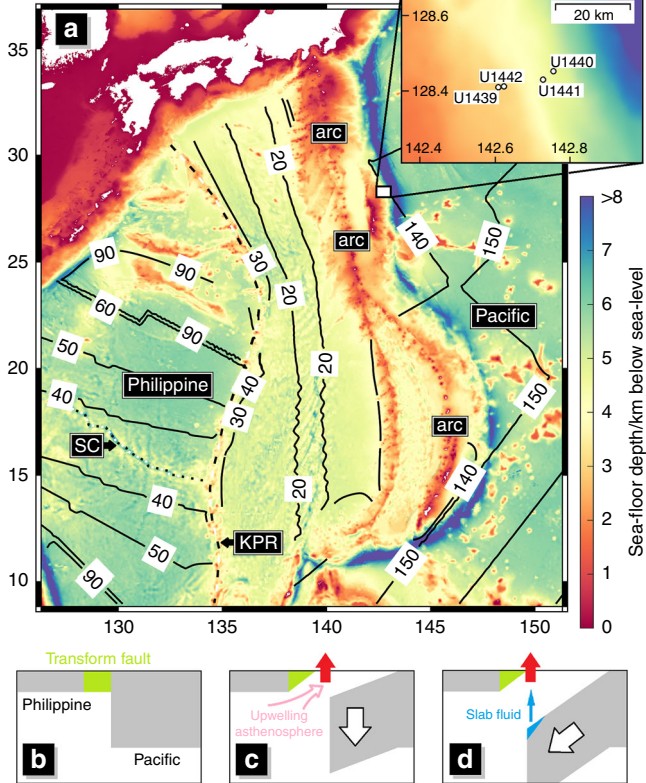

**Fig. 1 Region of study, sample locations and conceptual initiation model. a** Bathymetric map of the present day Izu-Bonin-Mariana (IBM) subduction system (created using the ETPOPO1 Global Relief Model of the United States National Oceanic and Atmospheric Administration) overlain by ocean floor age isochrones from Müller et al.[42] (solid black lines, labelled in Myr). The sites drilled by Expedition 352 are marked on the inset map. The ocean floor between the Kyushu-Palau ridge (KPR, marked by dashed black and white line) and the IBM arc has been formed by back-arc spreading since subduction initiated. The initial over-riding plate is therefore west of the KPR. In the south, an extinct spreading centre is visible (SC, marked by dotted black and white line) which would have been active at the time of subduction initiation. The volcanic arc, Philippine Sea and Pacific plates are also annotated. **b**, **c** The schematic model of subduction initiation driven vertically, by internal buoyancy differences, sometimes termed "spontaneous", previously proposed (Stern and Bloomer, 1992) to explain the magmatic record at the IBM. **b** is the initial condition with weak transform-fault material highlighted in green. **c** The initially older, denser, Pacific plate founders and asthenospheric material that wells up to fill the void melts via decompression to form fore-arc basalt. **d** Upwelling ceases as the slab begins to move along its length and the tip of the new slab begins to dehydrate/melt which fluxes the already depleted mantle above, causing it to melt further to form boninites.

## Results

**Modelling subduction initiation with variable forcing.** We set up a 2D numerical model of a 50 Myr old Pacific plate in contact

with a 5 Myr old Philippine Sea plate, the details of which are described in the methods section. In between the two plates is a 10 km wide pre-damaged (therefore plastically weak) transform zone. The over-riding plate would have been very young in the southern end of the IBM at the time, likely even actively spreading at a centre perpendicular to the transform, as can be seen by the isochrons west of the Kyushu-Palau ridge (Fig. 1). As we include a free surface in our model, a force similar to "ridge-push" is generated by the topographically higher, younger plate pushing against the older plate. For this initial condition, the magnitude of this internal push force is 2 TN/m (see methods). We apply an opposing and equal push force to the plates at the model boundary, such that horizontal forces are in balance at the start of the model run. A larger external force is applied when investigating the effect of far-field compression.

The two-dimensional models cannot capture the effects of along-strike propagation of subduction initiation. Three-dimensional models[19] show that subduction initiation can spontaneously start by vertical, buoyancy-driven, forcing (at for example a ridge-transform junction) and then progress by "un-zipping" along the transform fault; a process previously proposed by Stern and Bloomer[21]. External forcing (as proposed by Lallemand[22]), e.g. as a result of the Cenozoic plate reorganisation[11] or perhaps driven by vertical forces at the northern end of the transform fault, where Pacific plate subduction was already taking place at the Ryukyu subduction zone, could also trigger the propagation. We therefore allow for the application of a small additional vertical pull force to the Pacific plate in order to mimic out-of-plane forces along the boundary during such "un-zipping".

**Vertically driven subduction initiation and IBM style magmatism.** With the net initial horizontal push set to zero, our 2D representation of the initial condition of the Pacific-Philippine Sea plate system is already close to subduction initiation. We therefore find that only a small additional vertical pull force is required (9 TN/m, which is far less than a typical slab pull force of ~30 TN/m[23] suffices) for subduction to rapidly initiate and ultimately progress to fully down-dip subduction. Until now, this full evolution has not been observed in models of subduction initiation in which fore-arc spreading (necessary to form FABs, see below) also occurs.

During this process, the Pacific plate sinks into the mantle in such a way as to result in asthenospheric upwelling, and a lithospheric 'gap' forms beneath the old transform fault, just as in the conceptual model. The upwelling asthenosphere melts via decompression and we assume that these melts are fully extracted and erupt vertically above their point of formation, resulting in an average formation rate of FAB crust of ~4 cm/yr (Fig. 2a). As asthenosphere continues to flow into the widening gap (Fig. 2b), the erupted FABs are advected towards the sinking plate, where they accumulate in the region of the fore-arc closest to the future trench (Fig. 3a).

As the Pacific plate continues to sink, the crust at its tip crosses its solidus and begins to melt (Fig. 2c, in agreement with the Hf-Nd isotopic evidence for a slab melt contribution in the first erupted boninites[15]). At these shallow depths, the solidus coincides with the breakdown of the first hydrous minerals[24] and marks the onset of boninitic magmatism. Crucially, the time between the onset of FAB magmatism and the onset of boninitic magmatism is approximately 0.6 Myr (Fig. 2a, c), in striking agreement with drill core data[9]. The melting slab tip is spatially beneath the point where the FABs were previously erupted, and boninitic melts, once erupted, are also advected towards the subducting plate where they accumulate behind the FABs (nearer the future arc) (Fig. 3b). Indeed, the two drill holes from Exp. 352

nearest the trench were found to consist of FAB alone, and the two drill holes ~30 km closer to the arc consisted of only boninites[4]. Furthermore, our models predict that sediments are initially scraped off the slab, delaying their entry into the subduction zone, again in agreement with the isotopic identification of sediments in only the younger boninites of Exp. 352[15].

As subduction initiation proceeds, the sinking plate begins to move along its length. As it does so, asthenospheric material ceases flowing into the gap beneath the fore-arc and decompression melting moves further away from the trench towards the future sub-arc and back-arc region (Fig. 2d). The timescale of this potential back-arc magmatism is in agreement with IODP Exp. 351, which drilled west of the Kyushu-Palau ridge and recovered decompression melts in the back-arc region[25]. At this stage, new crust entering the nascent subduction system is no longer hot enough to melt, a mantle wedge with a normal corner flow is established, and as the new slab hydrates this convecting wedge, typical calc-alkaline magmatism will begin at the arc.

The evidence that FAB formation lasts for ~1 Myr along the ~2400 km length of the IBM fore-arc, coupled with the observation from previous 3D modelling that subduction initiation likely starts at one point and propagates along the transform fault[19], implies that subduction initiation of any one cross-sectional slice must have been extremely rapid. Indeed, the period of pre-boninite FAB generation in our 2D model (~0.6 Myr) is on the short end of the constraints from dating of the Exp. 352 drill cores (0.6–1.2 Myr)[9].

**Horizontally forced subduction initiation with no FAB.** We ran a suite of 20 models in which the external compressive force applied at the boundary was varied. We observed three types of behaviour in the models: no initiation, the "vertically driven" initiation described above and "horizontally forced" initiation (described below and in Fig. 2e–h). Figure 4 is a regime diagram that summarises the conditions required to achieve each type of behaviour. The boundaries in this diagram would shift if other input parameters that have been observed to affect transform fault collapse were changed. These include: the initial width of the fracture zone; the ages of either plate; the rheology of mantle and crustal material; the weakening effects of fluids[20,26]. However, while the boundaries between each type of behaviour in the regime diagram may move, the overall trends are likely to be robust.

Focusing on the behaviour of a model which displays "horizontally forced" initiation (specifically with 4 TN/m extra push, see Fig. 2e–h), no initial lithospheric gap is formed, shallow mantle upwelling with decompression melting does not occur, and thus no FAB is formed[12]. The tip of the subducting crust does cross its solidus however, so some magma with a "slab melt" signature might still be expected. In this primarily horizontally and externally forced case, it takes approximately 5 Myr for the Pacific plate to begin moving beneath the Philippine plate, but once it does so, subduction progresses at a similar speed to the vertically and internally driven case. This style of subduction initiation is likely occurring at the Puysegur ridge south of New Zealand[27] or in the Matthew and Hunter subduction system in the southwest Pacific[28]: much smaller systems in which initiation is more likely governed by external, horizontal forces.

## Discussion

As FAB is only formed in the vertically/buoyancy-driven type of subduction initiation in our models, the existence of FAB could be considered an indicator of this type of subduction initiation, in which forces internal to the initiating subduction system dominate over external forces. These internal forces could arise due to plate-age difference, such as in our example, or if one plate is

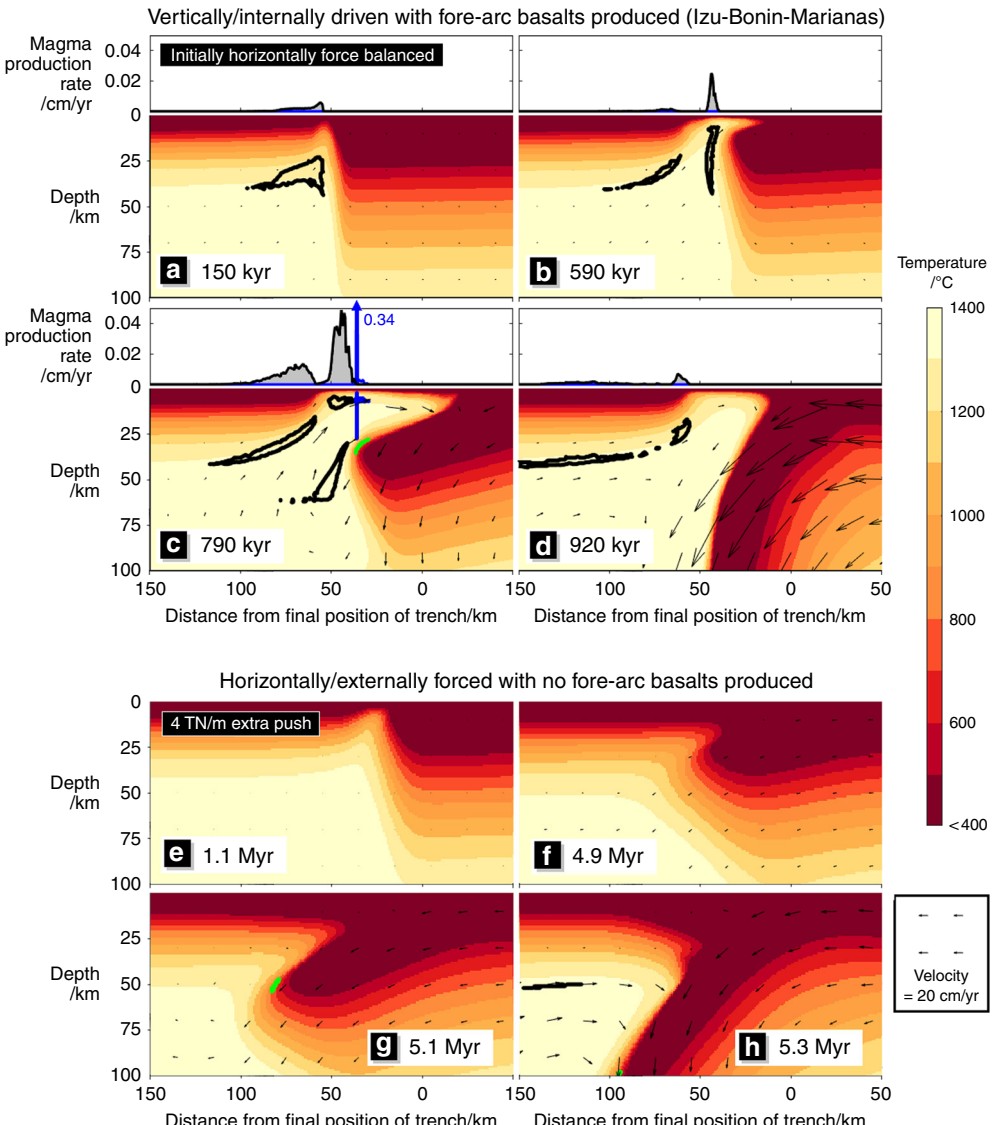

**Fig. 2 Models of vertically and horizontally driven subduction initiation.** Temperature profiles in **a–d**: the model displaying vertically driven subduction initiation (taken at 150, 590, 790 and 920 kYrs from the start of the model run); **e–h**: the model displaying horizontally forced subduction initiation (taken at 1.1, 4.9, 5.1, 5.3 Myr from the start of the model run). Overlain are: regions where decompression melting is occurring (black outline); where melting is occurring in the presence of slab fluid (blue outline) and the region of the subducting crust which has crossed its solidus (green outline). Above each profile (for the vertically driven model only, **a–d**) is the eruption rate of decompression melts, assuming that all melt is extracted and travels vertically to the surface.

comprised of old, buoyant arcs as has also been suggested[13]. The requirement that internal buoyancy forces have to dominate over external horizontal push likely means that such FAB-forming events are necessarily whole-plate scale (>1000 km) and therefore rare. However, when they happen, they would possibly be accompanied by more widespread changes in plate motions. Indeed, in this case, the initiation of the IBM just precedes the bend in the Hawaii Emperor seamount chain[29] (although whether IBM initiation caused or was catalysed this change in plate motion is debated[11]). The presence of FAB-like rocks in Tethyan ophiolites[30,31] may imply that subduction initiated along the Tethyan margin (stretching from the Alps to the Himalayas) in a similar self-driven, whole-plate scale manner.

## Methods

**Model setup.** We built a 2D model of a trench-perpendicular cross-section through the Izu-Bonin-Mariana subduction system (IBM) using the numerical

thermomechanical code Fluidity[32,33]. Fluidity solves the equations for conservation of momentum, mass, energy and composition using a finite element method on an adaptive triangular mesh which allows for a resolution of less than 300 m in regions containing either a high gradient in viscosity, boundaries between different compositions or melt generation (a resolution far finer than the length scales of the processes investigated in this study). The modelling domain is 1000 km wide and 300 km deep.

Initially, a 400 km long part of the Pacific and 600 km of the Philippine Sea plate are in direct contact (Fig. 5). The thermal profile of each plate is that of a half-space, cooled from above since their formation. The Pacific plate is 50 Myr old while the Philippine plate is 5 Myr old for the nearest 500 km to the Pacific plate then 50 Myr for the final 100 km, to act as an effective back-stop. The 10 km of the Philippine plate nearest the Pacific plate is compositionally "fracture zone" material from the surface, down to 40 km depth. These are reasonable dimensions for the damaged and highly hydrated zone around a mature fracture zone[34]. Other than this, the top 1 km of the model is "sediment", the 7 km below this is "crust" and the rest of the domain is "mantle". The only physical difference between compositions is the initial amount of "damage" (see below). The effect of changing many of these input parameters on the behaviour of this type of model has been explored by others[20,26,35]. We recognise that they may have an impact on the precise evolution of the model (and perhaps change the exact values of the applied forces needed in

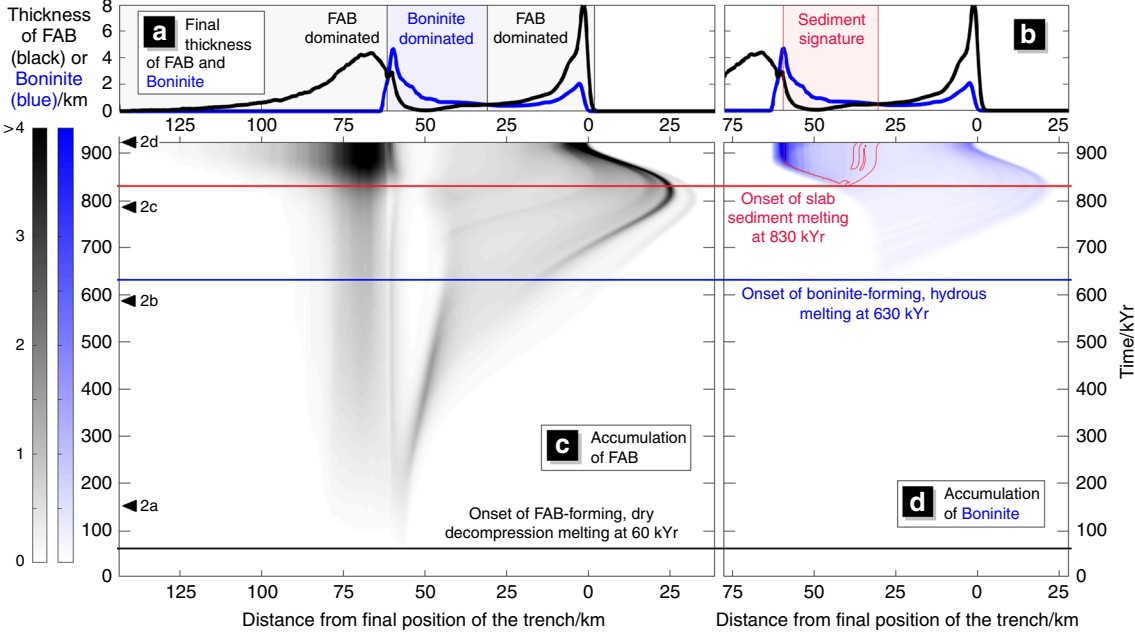

**Fig. 3 Modelled timeline of magmatism for vertically driven initiation.** Cumulative magma produced between the future trench and back-arc (*x*-axis) over time (*y*-axis). Magmatic products are advected with the velocity at the surface of the model. The intensity of the shading is the predicted thickness of magmatic product. Panel (**c**) is solely decompression melts that would become fore-arc basalt (FAB). Panel (**d**) is for melts produced in the presence of slab fluid which would become boninite. The area outlined in red in panel (**d**) is the region of the surface below which the subducting sediment has crossed its solidus. The markers "2a", "2b", "2c" and "2d" correspond to the times the snapshots in Fig. 2 (**a**, **b**, **c** and **d**) are taken. Above this, in panels (**a**) and (**b**), are the thicknesses of "FAB" (black line) and "boninite" (blue line) at the end of the model run. The region where sediment melts might be found is highlighted in red.

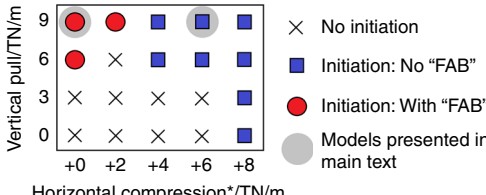

**Fig. 4 Subduction-initiation regime diagram.** The diagram summarises of all the models run during this study. The magnitudes of the vertical pull force, applied to the Pacific plate, and the external horizontal push force were varied. *Horizontal compression beyond what is needed to balance the initial, internally generated horizontal forces (see section S3). The two models described in more detail in the main text are highlighted with a grey circle.

order to observe each type of behaviour) and this is a potential avenue for further work.

**Boundary conditions and application of additional forces.** Prescribed pressure boundary conditions are chosen rather than prescribed velocity. As Leng and Gurnis[16] explain, velocity boundary conditions are only appropriate for modelling subduction initiation driven by far-field forces that are far larger than those generated locally and a large system like the IBM is more likely to be governed by internal rather than external forces. One important internally generated force is the horizontal push of the younger, more buoyant plate against the older plate. This is similar to "ridge-push" at mid-ocean ridges. In order for this force to arise naturally in our models, we include a free surface at the top of the model. The boundary at the base of the model is closed and non-slip. The Philippine-side boundary is closed and non-slip down to 80 km and open below, while the Pacific-side boundary is fully open (Fig. 5). An additional horizontal force is applied to the Pacific plate, implemented as an additional pressure applied at the boundary. The magnitude of this force is set to be equal to the magnitude of the force that the aforementioned "ridge-push" force that the Philippine Sea plate exerts on the Pacific plate at the fracture zone. It is calculated as follows:

$$F = g\rho_m \alpha (T_m - T_s)\left[1 + \frac{2\alpha(T_m - T_s)}{\pi}\right]\kappa(t_{paci} - t_{phil}), \quad (1)$$

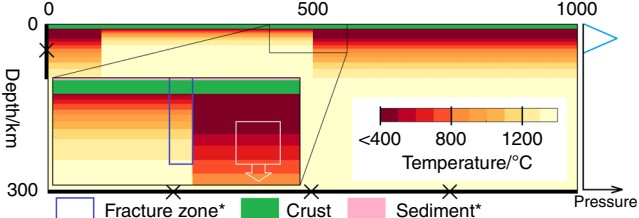

**Fig. 5 Schematic of the model setup and initial thermal condition.** Thermal boundary conditions are as the initial thermal condition (top boundary is at 0 °C and ambient mantle is 1350 °C). Mechanically, the top boundary is a free surface and the bottom closed and non-slip. The left (Philippine) side is closed and non-slip above 80 km and open below. The right (Pacific) side is open but an additional pressure can be added above 50 km to facilitate the application of an external push force. The schematic graph to the right of the figure is of the envelope of this applied pressure. Crust and sediment layers make up the top 7 and 1 km of the model domain initially and this material is tracked. The white box outlines the region to which the additional pull force is applied. *"fracture zone" material (outlined by the blue box) and sediment are initialised the maximum amount of "damage" (see main text).

where *g* is the acceleration due to gravity, $\rho_m$ is the mantle density, $\alpha$ is the thermal expansivity, $T_m$ and $T_s$ are the mantle and surface temperatures respectively, $\kappa$ is the thermal diffusivity and $t_{paci}$ and $t_{phil}$ are the ages of the Pacific and Philippine plates respectively (values of all constants in this equation, and all following equations, are given in Supplementary Table 1 in the supplementary material). This gives us a value of 2.0 TN/m given our particular model setup. The application of this force ensures that the horizontal forces within the model are initially balanced.

An additional pull force is applied to the Pacific plate within a region between 20 km and 40 km from the fracture zone and between 20 km and 40 km from the surface. This is achieved by increasing the density of material within this region. This pull force is a proxy for out-of-plane forces, assuming that subduction initiation starts at a different point along the transform fault to the 2D slice in our model[19,21].

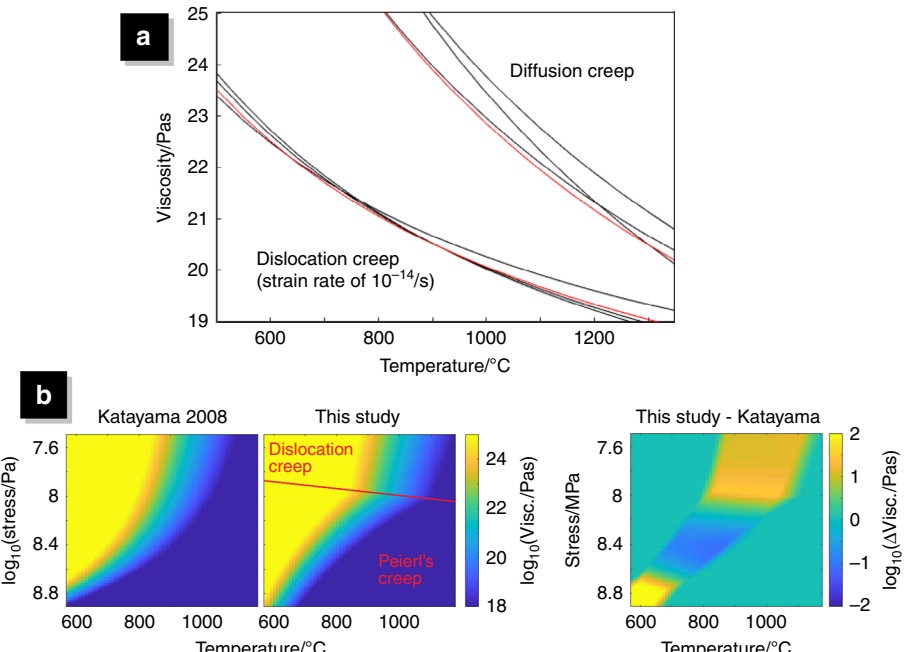

**Fig. 6 Illustration of method to choose flow law parameters. a** The flow laws used in this study, for dislocation creep and diffusion creep, are plotted in red. Experimentally determined flow laws for hydrated peridotite, commonly used for mantle material in geodynamic numerical models, are plotted in black for reference. The studies that determined these diffusion creep flow laws are (listed from strongest to weakest at 1000 °C): Hirth and Kohlstedt[43], Jaoul et al.[44], and Houlier et al.[45]. Similarly, the studies for these dislocation creep flow laws: Kirby and Kronenberg[46] and Hirth and Kohlstedt[43] Karato and Wu[47]. **b** We use an approximate Arrhenius formulation of the exponential flow law for Peierl's creep of hydrated peridotite, determined by Katayama and Karato[36] (as our modelling technique does not allow for exponential flow laws). We used a parameter grid search to minimise the total misfit between our flow law and the flow law of Katayama and Karato[36] in temperature-stress space (using our yield stress, or $8 \times 10^8$ Pa, as the maximum stress). These panels show the viscosity as given by Katayama and Karato[36]; the viscosity used in our study; and the minimised misfit.

**Rheology**. Material in the model deforms via the ductile mechanisms: diffusion, dislocation and Peierl's creep, as well as plastically. The effective viscosity due to ductile deformation, $\eta_{\mathrm{duct}}$, is calculated as follows:

$$\eta_{\mathrm{duct}} = \frac{1}{1/\eta_{df} + 1/\eta_{ds} + 1/\eta_{\mathrm{p}}}, \quad (2)$$

where $\eta_{df}$, $\eta_{ds}$ and $\eta_{\mathrm{p}}$ are the effective viscosities due to diffusion, dislocation and Peierl's creep which are calculated as follows:

$$\eta_i = A_i^{\frac{-1}{n_i}} \dot{\varepsilon}^{\frac{1-n_i}{n_i}} \exp\left(\frac{E_i + PV_i}{n_i RT}\right) i = \mathrm{df}, \mathrm{ds}, \mathrm{p}, \quad (3)$$

where $P$ is the pressure, $T$ is the temperature, $\dot{\varepsilon}$ is the second invariant of the strain-rate, $R$ is the gas constant and $A$, $E$, $V$ and $n$ are the rheological constants: the rheological prefactor, activation energy, activation volume and stress exponent, respectively. The values for diffusion and dislocation creep are set such that our flow law falls between three commonly used flow laws for dry peridotite, all determined by laboratory experiments (see Fig. 6a and caption). The values for Peierl's creep are set such that our Arrhenius formulation is the closest possible fit to the exponential flow law of Katayama and Karato[36] (see Fig. 6b and caption); determined by a parameter grid search.

Plastic deformation is approximated by using an effective viscosity, $\eta_{\mathrm{plas}}$, calculated by assuming a depth-dependent yield stress that follows a Byerlee law, as follows:

$$\eta_{\mathrm{plas}} = \frac{\min((C + \mu P), \tau_{\max})}{2\dot{\varepsilon}}, \quad (4)$$

where $\tau_{\max}$ is the yield stress, $C$ is the cohesion and $\mu$ is the friction coefficient, which is dependent on the total accumulated plastic train of the material, $\varepsilon_{\mathrm{p}}$. This method is a way of taking into account "damage" to the rock throughout the model run, which previous studies have shown is essential to achieve subduction initiation in models[37]. This is done in the following way:

$$\mu = \begin{cases} \mu_0 + (\mu_{\mathrm{f}} - \mu_0)\frac{\varepsilon_{\mathrm{p}}}{\varepsilon_{\mathrm{f}}}, & \varepsilon_{\mathrm{p}} < \varepsilon_{\mathrm{f}}, \\ \mu_{\mathrm{f}}, & \varepsilon_{\mathrm{p}} \geq \varepsilon_{\mathrm{f}}, \end{cases} \quad (5)$$

where $\varepsilon_{\mathrm{f}}$ is the strain at which material is considered maximally damaged, $\mu_0$ is the friction coefficient of undamaged material and $\mu_{\mathrm{f}}$ is the friction coefficient of maximally damaged material. We do not include "healing" as the timescales of the processes we are modelling are relatively short. Initially, the sediments and fracture

zone material are set to be maximally damaged and mantle material to have zero damage.

We combine ductile and plastic behaviour such that the final calculated viscosity, $\eta$, is

$$\eta = \frac{1}{1/\eta_{\mathrm{duct}} + 1/\eta_{\mathrm{plas}}}. \quad (6)$$

This is subject to capping at a minimum and a maximum viscosity of $10^{19}$ and $10^{24}$ Pa s, respectively.

**Calculating mantle melting**. We calculate mantle melting within the model by using the parameterisation of Katz et al.[38]. Details of how this is done can be found in the supplementary material and in this reference. Crucially, we assume that the mantle is hydrated by a constant amount of 0.05 bulk weight percent $H_2O$, when vertically above slab crust that is dehydrating and dry otherwise (this implicitly assumes that fluids only migrate vertically). At the shallow depths at which this occurs during this type of subduction initiation (<40 km), after the slab has lost its pore fluid in the vicinity of the trench (where it would not contribute towards melting), the next significant pulse of dehydration occurs at the crustal solidus[24]. This assumption is backed up in our case by the fact that even the first boninites produced appear to have a "slab melt" geochemical signature[15]. As such, we deem all super-solidus crust to be dehydrating and all sub-solidus crust to not be dehydrating. We choose to use the solidus of Bouilhol et al.[39]. for the slab crust, which lies very close to the experimentally derived solidus of Schmidt and Poli[40]. We also track where and when the sediment layer begins to melt, by using the solidus of Nichols et al.[41].

The rate of (fore-arc basalt forming) decompression melting, $\dot{F}$, is then given by

$$\dot{F} = \nabla F_{\mathrm{pot}} \cdot v, \quad (7)$$

where $v$ is the velocity field. The total amount of melting that material has undergone, or its depletion, is tracked and advected with the flow field. Mantle material only melts where the melting potential is higher than its depletion.

Once formed, all melt is assumed to be fully extracted and to travel vertically to the surface. We ultimately aim to predict how the different magmatic products are distributed by the time subduction is fully established. In order to achieve this, we advect the magmatic products according to the horizontal velocity at the top of the model during the model run, using an upwind finite difference scheme.

We end the model run when negligible decompression melting is occurring, and surface velocities are negligible in the fore-arc region. This happens before the

nascent slab nears the bottom boundary of our model and as such, the bottom boundary has little impact on the behaviours analysed in our models.

## Data availability

All model output is available from the corresponding author upon reasonable request.

## Code availability

The numerical code itself (Fluidity) is open source and available from https://fluidityproject.github.io/.

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

## Acknowledgements

The authors would like to acknowledge the International Ocean Discovery Program (IODP), staff and crew aboard the JOIDES Resolution during Exp. 352. Funding for B.M. and S.G. was provided as part of the Volatiles in the Lesser Antilles (VoiLA) project supported by the U.K. Natural Environment Research Council, NERC (grant NE/KO10743/1). Funding for shipboard participation in Exp 352 for J.P. was provided by NERC (NE/M010643/1) and for M.R. by NSF (OCE-1558647). This work used the ARCHER UK National Supercomputing Service (http://www.archer.ac.uk). The authors would also like to thank Rhodri Davies, Stephan Kramer and Tim Greaves for their support during model development and Jeroen van Hunen for valuable insight during correspondence.

## Author contributions

B.M. constructed, ran and processed the models and led writing of the manuscript with the help and guidance of S.G. J.P. proposed the original hypothesis and helped guide the study. J.P. and M.R. both provided expertise on the geochemistry and petrology of the region and aided in the writing of the manuscript.

## Competing interests
The authors declare no competing interests.
