## [Peer Review File · Nature Communications]

Reviewers' comments:

Reviewer #1 (Remarks to the Author):

The authors present two setups of a state-of-the-art geodynamic model of a well tested code with a formulation for damage rheology that is used to study initial plate sinking in the vicinity of a subduction initiation (SI) locus and relate the results to the detailed, recent, and uniquely complete observations at the IBM system. The model reproduces the temporal and spatial scales observed in the geologic record more accurately than previous models. The authors suggests that dominating internal forcing that arises due to topographic differences between subducting and overriding plates is the key to achieve this agreement between model and observation. The initiation of new subduction zones is a key unresolved aspect of the Earth Sciences, very important to understand, and the current study is very interesting and can proof useful to the whole Geoscience community.

The study, as it is presented now however, is based on a confusing and in some instances inconsistent set of words and concepts, which leads to a miscommunication and misrepresentation of the true nature of the underlying model. After the current manuscript has been thoroughly revised, I would see it as an important contribution to the current literature about the formation of new subduction zones, as one potential outcome of this study would allow us to link certain rock sequences to a certain SI dynamics.

GENERAL COMMENTS

Confusing wording

The use of terms without relation is a problem for this manuscript. The term „catastrophic“ (title; line 21; line 24; line 63; etc) is dominating this manuscript, yet it is not clear whether this is used in terms of temporal or spatial relation, or in comparison to other less catastrophic (?) SI events. It would make the manuscript, and its main point, much clearer if the authors would use a more specific term, or clearly specify what is meant with „catastrophic“. Other terms are „large scale“, and „rapid“ (see specific comments below). It is clear, for example, that individual initiation events cannot be significantly slower or faster (rapid) than others, when they are driven by internal forces (i.e., gravity mainly) only, as the authors assume here.

The authors argue that they reproduce what was once (confusingly) termed „spontaneous“ SI. Yet, the model prescribes plate convergence, or at least compression (via a stress boundary condition at the sides of the model - plus an „extra pull force“ to one of the plates), which would then according to the previous definition, clearly qualify as „forced“ SI - forced with external forces. This brings me to the next point of how the authors use „internal“ and „external“ forces. Far field tectonic forces (e.g., the ones applied to the model boundaries) are here termed „internal“ despite them being commonly considered „external“. The authors actually even flip between one and the other within the manuscript to describe the same force (e.g., lines 88-93 versus line 158).

I think the authors mix terms and concepts here, which causes confusion and results in an externally forced model of induced SI that is confusingly explained to the reader as being „spontaneous“ SI.

Also regarding Stern's terms „spontaneous“ and „induced“ SI, I think even the authors got confused by these terms to some degree and I therefore recommend re-evaluating whether to actually reuse these terms here, or not.

(B) Under- and misrepresentation of the model and no (mention of) model testing

The model assumes already ongoing subduction elsewhere along trench strike (Line 210-212), which was only clear after reading the methods section. This means the model is not really about subduction initiation, but about the induced (i.e., partially forced) sinking of a nearby and connected plate portion in the proximity of the actual subduction zone initiation event. This is

actually an interesting aspect and clearly worth modelling, but the model should probably also be communicated as such and not as a subduction initiation model, and, again, probably not as a "spontaneous subduction initiation" model.

The reader is left with four snapshots of the temperature field only and has to decipher the model behaviour from these two figures. Quantifying the velocity arrows or adding a velocity field plot would certainly be useful. For example, it seems that the „induced“ model with a larger induced compression is evolving much (i.e., almost an order of magnitude) slower than the „spontaneous“ model judging from the times given in both figures: Is that actually the case? It should be pointed out why.

The 300 km deep model base is closed AND non-slip (line 199) - why non-slip? - and therefore does not allow for material to flow out of the domain, nor easily along the bottom boundary? Does that influence the model in terms of slab sinking speed? Maybe this could be (or possibly has been) tested with additional models?

In addition the subducting plate is very short, meaning it has very little resistance to move towards the subduction zone compared to the Pacific plate on Earth at the time of the subduction onset. Is there anything in the model (which is not described yet) that accounts for a more natural, longer plate and the resulting flow resistance due to coupling of its base with the mantle?

A model setup figure (e.g. in the supplement) should be standard to any modelling study and could provide instructive information about all key forces occurring in and applied to the plate system.

The manuscript mentions just two models. Has the model been tested at all (numerical variations like e.g. resolution tests, or physical variations like e.g., model depth variations)? I think that would be highly important to do/mention.

(C) Reconstruction discussion and made assumptions

The authors argue that the IBM subduction zone initiation that external forces played a minor role in the IBM SI. Yet, several studies suggest that the IBM subduction zone initiated after/because of a regional plate reorganisation event in the West Pacific region, which might have occurred due to the subduction of the Izanagi-Pacific ridge beneath Asia at around 60-55 Ma (O'Connor et al., 2013; Lallemand, 2016) or the collision of the Olutorsky arc (Domeier et al., 2017) instead. This possibility should at least be discussed in relation to the key assumption of the current manuscript and to make the reader aware of potential external drivers and the difficulty to pinpoint an exact date of the SI event.

Domeier, M., Shephard, G. E., Jakob, J., Gaina, C., Doubrovine, P. V., & Torsvik, T. H. (2017). Intraoceanic subduction spanned the Pacific in the Late Cretaceous–Paleocene. *Science advances*, 3(11), eaao2303.

Lallemand, S. (2016). Philippine Sea Plate inception, evolution, and consumption with special emphasis on the early stages of Izu-Bonin-Mariana subduction. *Progress in Earth and Planetary Science*, 3(1), 15.

O'Connor, J. M., Steinberger, B., Regelous, M., Koppers, A. A., Wijbrans, J. R., Haase, K. M., ... & Garbe-Schönberg, D. (2013). Constraints on past plate and mantle motion from new ages for the Hawaiian-Emperor Seamount Chain. *Geochemistry, Geophysics, Geosystems*, 14(10), 4564-4584.

Also, it is suggested that the Proto-Philippine Sea plate (i.e., the upper plate during IBM SI) was mostly formed of arc terranes at the time of subduction initiation (e.g., Ishizuka et al., 2018). Could/should that be considered in the modelling?

SPECIFIC COMMENTS

line 11: Would the reference for this statement (as given later on in the text) be useful to the reader already at this point?

line 14: The authors could consider clarifying „rapid“, which currently stands on its own without any actual time relation. Rapid compared to other sequences of subduction initiation (are there other examples?), or rapid compared to the geologic time scale, or something else?

line 20; line 24: Again, what does „large-scale“ mean in this context?

line 46: It is not clear to me what the authors mean here by „regional scale“? Is there „global scale“ SI?

line 51: Consider using „has been found“ instead, to indicate that there might be more similarly complete magmatic records of SI yet to be found.

line 63: terms without relation: „rapid“; „catastrophic“.

lines 64-66: This (i.e., SI inducing plate reorganisation) might be true, but it could also be the other way round, which might be pointed out to the reader: see main comment above.

line 88: the authors should consider clarifying that what has been pointed out is really that history dependent plasticity is important for SI „in models“, but not necessarily in general.

lines 88-93: Oh, so far field plate motion is considered here being an internal force? Shouldn't the external tectonic forcing (which is applied to the side boundaries of the model) be considered an external force instead? In my opinion, such forcing is generally considered external. In any case, this needs to be clarified (see also my main comment on external forces).

lines 92-93: The authors say here that the push solely arises from topographic variations of the upper plate, but stated two lines before that they impose plate compression (i.e., a push) on the sides of the model domain. This needs to be clarified.

lines 92-93 again: A free surface (i.e., surface topography) only causes a horizontal push, if the upper plate varies in thickness laterally. Is this actually the case; from what I can tell (e.g., Fig. 2) it is not? Please clarify.

line 107: specify what you mean with „internal push force“. Is it the externally imposed model boundary condition or the compressive force arising from what the authors described as topography driven?

line 109-110: some words missing.

lines 109-110: OK, this „extra force“ to induce SI is important, when the study is about SI. Please clarify: e.g., Where is the „pull“ applied and to what direction? Where does this force come from in a natural example? Arguing with it being smaller than a typical slab pull force does not make sense here, since the authors study the onset of subduction, which by definition has no pre-existing slab and therefore zero slab pull.

Line 112: Here saying that the IBM subduction started as a point source (i.e., in one location in space) makes sense, but now appears conflicting the rest of the manuscript arguing for a "catastrophic" event that should have occurred along a wide trench segment? Do I misunderstand this? Please clarify.

Lines 115-117: This sentence, as formulated now, is not true: Subduction initiation has been

modelled already previously even without any imposed boundary conditions (e.g., Crameri and Tackley 2016 or Davaille et al. 2017).

Crameri, F., and P. J. Tackley (2016), Subduction initiation from a stagnant lid and global overturn: new insights from numerical models with a free surface, *Progress in Earth and Planetary Science*, 3(1), 1–19, doi:10.1186/s40645-016-0103-8

Davaille, A., Smrekar, S.E., Tomlinson, S., 2017. Experimental and observational evidence for plume-induced subduction on Venus. *Nat. Geosci.* <http://dx.doi.org/10.1038/ngeo2928>.

Line 142: Use Myr instead of Myrs.

Line 158: Specify what actually changes (as subduction initiation remains subduction initiation), e.g., the SI dynamics, or the SI timing, etc.

Line 158: The compressive force is now an "external force"? I think that is correct, but the authors claimed elsewhere that these forces are internal instead (see main comment).

Line 164: Duarte's model for SI does not make a good example here, as he argued, I think with delamination being the main cause for subduction to initiate off coast Portugal and not compression. Where would such a compression come from anyway there?

Line 166: This sentence appears problematic, since the authors don't make it clear what external and internal forces are exactly. In addition, making such a statement based on what, two models, seems a bit overly confident.

Line 170: The term "large scale" - even when given with a length scale - needs to be specified somewhere. Why not just saying something like "...such events are inducing a new subduction trench simultaneously along a xxx km wide segment". I think that would make it much clearer.

Line 174: I don't think what the authors showed here in their models was actually "self-driven", as they imposed external compression via side boundary conditions.

Line 192 and other instances: Use "Myr" and not "Myrs"

Line 210-212: So, this is actually a model for subduction progression, not initiation? I think that is fine and useful, but probably should be clarified throughout the main part of the manuscript.

Figure 2:

The abbreviation for „years“, or „annum“ is somewhat undefined in the Geosciences. Yet, „Yrs“ (i.e., a plural form) is certainly not a dimension. Consider using one of the other commonly used dimensions, e.g., „y“ or „yr“ (i.e., „ky“ or „kyr“) for time intervals, as the authors actually already do elsewhere (e.g., line 103).

(also Figure S2) From the figure it appears that the surface temperature is 400 degree C. Is that actually the case?

Table S1: Use „Myr“ here too.

Figure S2: What are the green (panel c) and black (panel d) solid lines representing?

Fabio Crameri
07.08.2019

Reviewer #2 (Remarks to the Author):

This is an interesting and timely paper discussing dynamics of subduction initiation and associated magmatism on the basis of 2D numerical models. Modeling results are compared to observations from IBM subduction system. It is demonstrated that only spontaneous (i.e., weakly/moderately forced) subduction initiation across a transform fault is capable to reproduce the IBM magmatic record (FAB=>boninites) whereas induced (i.e., strongly forced) subduction initiation scenario lacks the early decompression melting phase that is necessary to produce FAB. The paper is well written and is of broad interest but needs some reorganization and improvements:

1. Fig. S2 should be moved to the main text since it is as important as Fig. 2. Both "spontaneous" and "induced" reference models should be described in the main text to illustrate the major conclusion.

2. Sensitivity of the modeling results to variations in model parameters (initial model geometry and thermal structure, rheology, faults width etc.) should be described. All models run for this paper needs to be described in the supplement (a Table with models parameters and results). Otherwise the paper creates an impression that only two shown models were run that is definitely insufficient in light of previous systematic studies (e.g., Hall et al., 2003; Gerya et al., 2008; Dymkova and Gerya, 2013, Nikolaeva et al., 2008; Zhu et al., 2009, 2011 etc.) that showed strong sensitivity of results to various model parameters.

3. Previous numerical modeling literature overview needs some improvements since some of the claims made for previous studies do not seem entirely correct.

Specific comments on the paper are given below.

Taras Gerya, 15.08.2019 Zurich

Specific comments

Lines 77-80. "While this conceptual model has been partially tested dynamically (Dymkova and Gerya, 2013; Leng and Gurnis, 2011; Leng et al., 2012; Zhou et al., 2018), none reproduce the rapid time scale that the recent IBM data require, nor have they been shown to produce the very distinct magmatic sequence that Exp. 352 recovered." It would be good to give more details concerning the timescales. Transform collapse time scales can be very rapid (few Myr or less, e.g. Gerya et al., 2008; Gerya, 2019, Chapter 17 or Gerya, 2019, Chapter 21, subduction example). Magmatic record was to some degree explored by Nikolaeva et al. (2008) and Zhu et al. (2009; 2011), however for fluid fluxed melting only (without decompression melting).

Lines 88-90. "We chose to use stress boundary conditions rather than a prescribed velocity as behaviour in such a large system is more likely to be governed by internal rather than external forces." Strictly saying, in case of imposed external stresses, subduction initiation regime is not purely spontaneous, since initiation is driven both by local buoyancy contrast and external forcing.

Figure 2: "Temperature profiles taken at 150, 590, 790 and 920 kYrs from the start of the model run." Rapid slab dynamics shown on this figure seems similar to these in previous studies (e.g. Gerya et al., 2008; Gerya, 2019, Chapter 17 or Gerya, 2019, Chapter 21, subduction example; Nikolaeva et al., 2008; Zhu et al., 2009; 2011; Zhou et al., 2018).

Lines 141-142 "Crucially, the time between on the onset of FAB magmatism and the onset of boninitic magmatism is approximately 0.5 Myrs (Fig. 2a,c), in perfect agreement with drill core data (Reagan et al., 2019)." How does this time depend on model parameters (rheology, thermal/compositional structure etc.). The dynamics of the initiation is strongly sensitive to these (e.g., Hall et al., 2003; Gerya et al., 2008; Dymkova and Gerya, 2013)

Supplement. Lines 100-103 "At the relatively shallow depths that we are interested in, the crustal solidus is the most significant pulse of water loss from the slab, after the early loss of pore fluid (van Keken et al., 2011), and we assume here that slab fluids migrate vertically upwards." What about serpentine dehydration? How fast does the fluid move? Is this motion affected by the mantle

flow as in previous models (e.g., Gerya et al., 2008; Nikolaeva et al., 2008; Zhu et al., 2009, 2011)

References

- Dymkova, D., Gerya, T. (2013) Porous fluid flow enables oceanic subduction initiation on Earth. *Geophysical Research Letters*, 40, 5671–5676.
- Gerya T.V. (2010) *Introduction to Numerical Geodynamic Modelling*. Cambridge University Press, 345 pp.
- Gerya T.V. (2019) *Introduction to Numerical Geodynamic Modelling*. Second Edition. Cambridge University Press, 472 pp.
- Gerya, T.V., Connolly, J.A.D., Yuen, D.A. (2008) Why is terrestrial subduction one-sided? *Geology*, 36(1), 43-46.
- Hall, C.E., Gurnis, M., Sdrolias, M., Lavier, L.L., Muller, R.D., 2003. Catastrophic initiation of subduction following forced convergence across fractures zones. *Earth Planet. Sci. Lett.* 212, 15–30.
- Nikolaeva, K., Gerya, T.V., Connolly, J.A.D. (2008) Numerical modelling of crustal growth in intraoceanic volcanic arcs. *Phys. Earth Planet. Interiors*, 171, 336-356.
- Zhou, X., Li, Z.H., Gerya, T.V., Stern, R.J., Xu, Z.Q., Zhang, J.J. (2018) Subduction initiation dynamics along a transform fault control trench curvature and ophiolite ages. *Geology*, 46, 607-610
- Zhu, G., Gerya, T.V., Yuen, D.A., Honda, S., Yoshida, T., Connolly, J.A.D. (2009) 3-D Dynamics of hydrous thermalchemical plumes in oceanic subduction zones. *Geochemistry, Geophysics, Geosystems*, 10, Article Number: Q11006.
- Zhu, G., Gerya, T.V., Honda, S., Tackley, P.J., Yuen., D.A. (2011) Influences of the buoyancy of partially molten rock on 3-D plume patterns and melt productivity above retreating slabs. *Phys. Earth Planet. Interiors*, 185, 112-121.

Reviewer #3 (Remarks to the Author):

Review of the Manuscript #NCOMMS-19-25654-T, entitled "Dynamics of catastrophic plate boundary formation in the western Pacific", by Maunder et al.

In this manuscript, the authors investigate the problem of subduction initiation using advanced numerical models coupled with natural observations from the long Izu-Bonin-Mariana subduction system. The forearc of this subduction zone has been recently drilled by the IODP and has been a key site for our understanding of how subduction zones initiate. Subduction initiation is a hot topic in Earth sciences. We still do not understand well how subduction zones initiate, and since subduction zones are the main driver of plate tectonics and mantle convection this understanding is crucial for our understanding of the Earth system.

I understand that the authors present arguments for spontaneous subduction initiation along a pre-existent weak (fracture) zone and suggest that such mode of subduction initiation is required to explain the observations. Alternative models have proposed that most of the subduction systems are forced by external forcing. Their proposal seems to be based on robust assumptions, observations and modelling results. I believe this will be a strong contribution to the discussion and will cause some controversy (in a positive sense), which I think is essential to keep the discussion on this topic going and further improve our knowledge. Therefore, the impact on the community is expected to be high.

The manuscript is very well written and the science is of top quality. I really enjoyed reading this paper and I find it a timely contribution. I, therefore, recommend it for publication in *Nature*

Communications.

Minor comments:

- Please consider including the term "subduction initiation" in the title instead of "plate boundary". I believe this may increase the impact of the paper as people will immediately know that it deals with subduction initiation.

- Line 141: typo "on"

- Line 142: avoid terms such as "perfect". They are too subjective.

Joao Duarte
IDL, University of Lisbon

Responses to reviewers' comments

We would like to thank the reviewers Fabio Cramer, Taras Gerya and Joao Duarte for their fair and constructive comments. By taking on board their suggestions and addressing their concerns, we have been able to produce a manuscript that we believe is both more rigorous and far clearer.

One of the main points raised by the reviewers focused around our confusing use of terminology throughout the manuscript. We have addressed this by doing away with the terms “spontaneous” and “induced” and instead using the terms “vertically”, “internally” or “buoyancy” driven where appropriate and “horizontally” or “externally” forced, when referring to the two types of model behaviour observed in this study. These terms are presently being adopted throughout the community and as such, by making this change, we align our work better with other work currently being published on the topic. We have also been particularly careful to emphasise what we mean by “internal forces”: whether we mean forces that are internal to our 2D model or forces that are internal to the full 3D subduction system as a whole. Other terms highlighted as being problematic by the reviewers (e.g. “large-scale” and “catastrophic”) have been replaced, clarified or justified on a case-by-case basis.

Another key point raised by the reviewers was the lack of description of the full suite of models that we ran for this study. We have now included a section in the supplementary material which summarises the key results from the full model suite and refer to the full model suite more regularly throughout the main text.

We have also heavily edited the introduction and discussion on recommendation by the reviewers. Among the added content is a more accurate description of previous modelling work done and a discussion of alternative arguments with regards to the relationship between IBM initiation and Cenozoic plate re-organisation.

Please find below our responses to the comments of the three reviewers on our manuscript, originally titled “Dynamics of catastrophic plate boundary formation in the western Pacific” but now retitled “Dynamics and magmatism of rapid plate subduction initiation in the western Pacific”. The original comments are in black and our responses are in blue.

Reviewer #1 (Remarks to the Author):

The authors present two setups of a state-of-the-art geodynamic model of a well tested code with a formulation for damage rheology that is used to study initial plate sinking in the vicinity of a subduction initiation (SI) locus and relate the results to the detailed, recent, and uniquely complete observations at the IBM system. The model reproduces the temporal and spatial scales observed in the geologic record more accurately than previous models. The authors suggests that dominating internal forcing that arises due to topographic differences between subducting and overriding plates is the key to achieve this agreement between model and observation.

The initiation of new subduction zones is a key unresolved aspect of the Earth Sciences, very important to understand, and the current study is very interesting and can proof useful to the whole Geoscience community.

The study, as it is presented now however, is based on a confusing and in some instances inconsistent set of words and concepts, which leads to a miscommunication and misrepresentation of the true nature of the underlying model. After the current manuscript has been thoroughly revised, I would see it as an important contribution to the current literature about the formation of new subduction zones, as one potential outcome of this study would allow us to link certain rock sequences to a certain SI dynamics.

GENERAL COMMENTS

Confusing wording

The use of terms without relation is a problem for this manuscript. The term „catastrophic“ (title; line 21; line 24; line 63; etc) is dominating this manuscript, yet it is not clear whether this is used in terms of temporal or spatial relation, or in comparison to other less catastrophic (?) SI events. It would make the manuscript, and its main point, much clearer if the authors would use a more specific term, or clearly specify what is meant with „catastrophic“. Other terms are „large scale“, and „rapid“ (see specific comments below). It is clear, for example, that individual initiation events cannot be significantly slower or faster (rapid) than others, when they are driven by internal forces (i.e., gravity mainly) only, as the authors assume here.

The authors argue that they reproduce what was once (confusingly) termed „spontaneous“ SI. Yet, the model prescribes plate convergence, or at least compression (via a stress boundary condition at the sides of the model - plus an „extra pull force“ to one of the plates), which would then according to the previous definition, clearly qualify as „forced“ SI - forced with external forces. This brings me to the next point of how the authors use „internal“ and „external“ forces. Far field tectonic forces (e.g., the ones applied to the model boundaries) are here termed „internal“ despite them being commonly considered „external“. The authors actually even flip between one and the other within the manuscript to describe the same force (e.g., lines 88-93 versus line 158).

I think the authors mix terms and concepts here, which causes confusion and results in an externally forced model of induced SI that is confusingly explained to the reader as being „spontaneous“ SI. Also regarding Stern's terms „spontaneous“ and „induced“ SI, I think even the authors got confused

by these terms to some degree and I therefore recommend re-evaluating whether to actually reuse these terms here, or not.

The reviewer has a very valid point here that the terms “spontaneous” and “induced” are misleading. We use the term “spontaneous” to refer to subduction initiation which progresses in the way proposed by Stern and Bloomer 1992. We made this decision so that our manuscript used the same terminology as previous literature, in which the term spontaneous has become shorthand for this conceptual model.

Indeed, in our model our “spontaneous” initiation isn’t spontaneous in the strictest sense: we apply both a small horizontal push and vertical pull as the reviewer rightly points out. However, we would argue that subduction subsequently progresses essentially spontaneously, in that forces that are internal to the system as a whole (i.e. buoyancy forces, both in-plane and out of plane of our 2D model: introduced via the application of the small additional slab pull force, something we now describe more thoroughly in the manuscript) must dominate the system for the model to progress via a Stern-type mechanism and for FAB to be produced. As soon as (external) horizontal push forces increase to a value significantly greater than what is needed to balance the internally generated horizontal forces, FAB generation no longer occurs.

In our manuscript we previously argued that we are modelling a cross section through the IBM away from the point at which a small disequilibrium started in the initial force balance that leads to the progression to subduction initiation (SI). The inception of disequilibrium at this initial point need not be spontaneous either: it could be due to stress localisation at some point along the transform fault or subduction could have even been inherited from one end (the Northern end perhaps where Pacific plate subduction was already taking place at the Ryukyu trench). Plate reorganisation may have even been the initial catalyst (see answer to a later point). We now discuss these worthwhile points in the main text. Important is that once thus started, the internal forces in the system are sufficient to lead to rapid SI accompanied by the generation of FABs and then boninites.

We did already try to convey these points in the manuscript but our choice of terminology was clearly not helpful. We have therefore decided to describe the two modes of subduction initiation as “vertically” or “buoyancy” driven and “horizontally” or “externally” forced. Indeed, in correspondence with some of the authors of a submitted Subduction Initiation review paper (of which the reviewer is the lead author), we discussed how the terms “vertically” and “horizontally” induced are less confusing: a key point of this review paper. Therefore, by also employing these terms, the paper is now in line with this latest framework. At the beginning of the manuscript we now clearly define these terms and how they relate to “spontaneous” and “induced” for easier comparison with previous work.

We also can see that a source of confusion is that forces internal to our 2D model and forces internal to the IBM system as a whole are two different things. We are more careful to differentiate between the two.

(B) Under- and misrepresentation of the model and no (mention of) model testing

The model assumes already ongoing subduction elsewhere along trench strike (Line 210-212), which

was only clear after reading the methods section. This means the model is not really about subduction initiation, but about the induced (i.e., partially forced) sinking of a nearby and connected plate portion in the proximity of the actual subduction zone initiation event. This is actually an interesting aspect and clearly worth modelling, but the model should probably also be communicated as such and not as a subduction initiation model, and, again, probably not as a “spontaneous subduction initiation” model.

The reviewer is correct that the model requires an additional vertical pull force in order for subduction to initiate and the most likely source for this force is from a subduction section that is already in the process of initiating; i.e. we are modelling the “unzipping” of a subduction zone as the reviewer points out. We have made this clearer earlier in the manuscript. However, we would argue that this is still the initiation of subduction at the IBM. We have included more discussion and literature review focussing on what led to this “unzipping” in the first instance.

We also describe better the wider model suite that we ran and include a section in the supplementary material which summarises the key results from the full parameter study. Please see the answer to this specific point below.

The reader is left with four snapshots of the temperature field only and has to decipher the model behaviour from these two figures. Quantifying the velocity arrows or adding a velocity field plot would certainly be useful.

We have added a scale for the velocity arrows in Figure 2.

For example, it seems that the „induced“ model with a larger induced compression is evolving much (i.e., almost an order of magnitude) slower than the „spontaneous“ model judging from the times given in both figures: Is that actually the case? It should be pointed out why.

To address a point raised by reviewer 2 we have combined Figure 2 and Figure S2 and included more discussion directly comparing the two models and explaining their differences which also addresses this point.

The 300 km deep model base is closed AND non-slip (line 199) - why non-slip? - and therefore does not allow for material to flow out of the domain, nor easily along the bottom boundary? Does that influence the model in terms of slab sinking speed? Maybe this could be (or possibly has been) tested with additional models?

By using this boundary condition we are simply assuming that the mantle below 300 km is static and that we are in its reference frame. All analysis done on the model is done while the slab is above 150 km and far from the bottom boundary, near which there is no significant mantle flow throughout the model run.

In addition the subducting plate is very short, meaning it has very little resistance to move towards the subduction zone compared to the Pacific plate on Earth at the time of the subduction onset. Is

there anything in the model (which is not described yet) that accounts for a more natural, longer plate and the resulting flow resistance due to coupling of its base with the mantle?

The reviewer points out a model limitation here which we also recognise. All forces outside the modelling domain have to be approximated by our mechanical boundary conditions, this includes plate drag. It stands to reason therefore that if we were to change the size of the box, say increase it, we may find that we have to slightly adjust the forces we apply at the boundaries to achieve similar behaviour. However, we do not believe that this weakens our conclusions.

A model setup figure (e.g. in the supplement) should be standard to any modelling study and could provide instructive information about all key forces occurring in and applied to the plate system.

We have now included a model setup figure in the supplementary material: Figure S1.

The manuscript mentions just two models. Has the model been tested at all (numerical variations like e.g. resolution tests, or physical variations like e.g., model depth variations)? I think that would be highly important to do/mention.

Resolution in our model is adaptive and, within the regions of interest (where melting occurs or where high velocity gradients exist), resolution is extremely fine by the standards of numerical geodynamic models (< 300 m). We did not run computationally demanding tests with increased resolution as the resolution is already finer than the length scales of the processes we are investigating, and all the techniques we are using are well tested.

We did however run a suite of models varying the key parameters in this study: the vertical pull force and the horizontal push force. We recognise the importance of making the results of this full parameter study available to the reader but would argue that it is not important to the main conclusions of this paper. As such, we have included a summary of the full parameter exploration in the supplementary material.

We did not vary other parameters, such as transform fault width, rheology, plate ages etc. Instead we gave all other model properties best-estimate values which were carefully informed by the current literature. This is already outlined in the methods and supplementary material. We recognise that they will likely have an effect on the exact behaviour within the model, (specifically perhaps changing the exact values of the applied forces needed in order to observe each type of behaviour); however previous work has already been done on the effects of many of these parameters on the system that we are modelling here. Indeed, this work was considered when choosing values for these parameters. In supplementary section S5, where we summarise the results from the full model suite, we also raise and elaborate on this point to make all of the above clear to the reader.

Please see the replies to the specific points below with regards to the effect of both horizontal and vertical box size.

(C) Reconstruction discussion and made assumptions

The authors argue that the IBM subduction zone initiation that external forces played a minor role in

the IBM SI. Yet, several studies suggest that the IBM subduction zone initiated after/because of a regional plate reorganisation event in the West Pacific region, which might have occurred due to the subduction of the Izanagi-Pacific ridge beneath Asia at around 60-55 Ma (O'Connor et al., 2013; Lallemand, 2016) or the collision of the Olutorky arc (Domeier et al., 2017) instead. This possibility should at least be discussed in relation to the key assumption of the current manuscript and to make the reader aware of potential external drivers and the difficulty to pinpoint an exact date of the SI event.

O'Connor et al. 2013 argue that the Hawaii-Emperor Seamount Chain bends due to the onset of subduction of the Marianas-Tonga system. They cite Whittaker et al. 2007 who suggest that the subduction of the Pacific-Izanagi ridge caused the change in plate motion which led to compression across the initial fracture zone. Lallemand 2016 runs with this idea, coming up with an alternative model to explain the presence of FAB in the IBM forearc. However, his model doesn't seem to explain the narrow age-range of FAB (1-2 Myr) ages along the entire length of the IBM. Our model neatly explains this observation, along with the occurrence of FAB in the Western Philippine basin, the presence of which Lallemand used as evidence against the Stern model of subduction initiation in the IBM.

However, we do not wish to rule out IBM initiation being catalysed somehow by plate reorganisation either and this has been added to the discussion.

Also, it is suggested that the Proto-Philippine Sea plate (i.e., the upper plate during IBM SI) was mostly formed of arc terranes at the time of subduction initiation (e.g., Ishizuka et al., 2018). Could/should that be considered in the modelling?

We do actually already discuss the fact that this could be an additional source of vertical forcing (via an additional buoyancy contrast between the two plates) in the final paragraph of the main text. We did not explicitly investigate the effect of an arc terrane to keep the study simple, although we do agree with the reviewer that it is important to mention.

SPECIFIC COMMENTS

line 11: Would the reference for this statement (as given later on in the text) be useful to the reader already at this point?

We have added a reference to Forsyth and Uyeda, 1975 here.

line 14: The authors could consider clarifying „rapid“, which currently stands on its own without any actual time relation. Rapid compared to other sequences of subduction initiation (are there other examples?), or rapid compared to the geologic time scale, or something else?

We simply mean rapid for such a tectonic process and in comparison with similar tectonic processes of a similar spatial scale. The timescale is quantified when used here and then this elaboration is added when first used in the introductory text at line XX.

line 20; line 24: Again, what does „large-scale“ mean in this context?

We have reworded “large-scale” to “whole-plate scale” as this is more specific and better illustrates what we mean: large enough for internal forces to be dominant.

line 46: It is not clear to me what the authors mean here by „regional scale“? Is there „global scale“ SI?

We were actually using this to mean “as opposed to very local scale” rather than “opposed to global scale” but we can definitely see how this could be confusing and again we replace this with “whole-plate scale” which is more specific.

line 51: Consider using „has been found“ instead, to indicate that there might be more similarly complete magmatic records of SI yet to be found.

This has been changed.

line 63: terms without relation: „rapid“; „catastrophic“.

This is where we have added clarification for “rapid”, and “catastrophic” has been removed. We have also edited the use of “large scale” in the following sentence.

lines 64-66: This (i.e., SI inducing plate reorganisation) might be true, but it could also be the other way round, which might be pointed out to the reader: see main comment above.

We have extended this point to mention this idea using Whittaker et al. 2007 as a reference.

line 88: the authors should consider clarifying that what has been pointed out is really that history dependent plasticity is important for SI „in models“, but not necessarily in general.

“in models” has been added to the end of this sentence.

lines 88-93: Oh, so far field plate motion is considered here being an internal force? Shouldn't the external tectonic forcing (which is applied to the side boundaries of the model) be considered an external force instead? In my opinion, such forcing is generally considered external. In any case, this needs to be clarified (see also my main comment on external forces).

Again, we can see that the use of “internal forces” to refer to both forces internal to the 2D model and forces internal to the whole IBM system is confusing. See answer to the main point above. However, the force being referred to in these sentences is truly internally generated within the model so we keep this description here, adding clarity by changing the wording.

lines 92-93: The authors say here that the push solely arises from topographic variations of the

upper plate, but stated two lines before that they impose plate compression (i.e., a push) on the sides of the model domain. This needs to be clarified.

The reviewer refers to a statement two lines before where we describe our choice of mechanical boundary conditions. We are not introducing the horizontal push force here and have changed the wording to make this clearer. The internally generated push force described here truly does arise from just topographic variations. In the next paragraph we go on to describe how we then choose to exactly balance this internal force with an imposed external push so that the system begins in an initially horizontally force balanced state. We have also reworded this part for clarity.

lines 92-93 again: A free surface (i.e., surface topography) only causes a horizontal push, if the upper plate varies in thickness laterally. Is this actually the case; from what I can tell (e.g., Fig. 2) it is not? Please clarify.

The lateral variation is in the initial condition at the transform fault. The fact that the younger plate sits higher than the older plate means that the pressure beneath the younger plate is higher (until a depth at which the mantle is the same temperature on both sides). Such a horizontal pressure gradient results in a push force between the two plates. We now direct the reader to the supplementary material at this point in the text as section S3 outlines how this force is calculated and should add clarification for any reader who is unsure of this point.

line 107: specify what you mean with „internal push force“. Is it the externally imposed model boundary condition or the compressive force arising from what the authors described as topography driven?

We are referring to the argument that stress boundary conditions are an appropriate choice for a system in which internal horizontal forces matter and velocity boundary conditions more appropriate for a system which is being driven by external forces (e.g. at a fixed velocity). This is an argument actually introduced and elaborated upon by Leng et al. 2011. We reword this sentence and include this reference here for extra clarity.

line 109-110: some words missing.

The word “of” is added.

lines 109-110: OK, this „extra force“ to induce SI is important, when the study is about SI. Please clarify: e.g., Where is the „pull“ applied and to what direction? Where does this force come from in a natural example? Arguing with it being smaller than a typical slab pull force does not make sense here, since the authors study the onset of subduction, which by definition has no pre-existing slab and therefore zero slab pull.

We can see that the application of the extra pull force needs a clearer introduction, explanation and justification. We have rewritten this section to include more detail and reordered it so that the intention of the force representing out-of-plane forces at the subduction zone “unzips” is clearer. It is also made clear to the reader that we are not addressing the question of how the unzipping

started. We do however summarise the current suggestions currently given in the literature.

Line 112: Here saying that the IBM subduction started as a point source (i.e., in one location in space) makes sense, but now appears conflicting the rest of the manuscript arguing for a "catastrophic" event that should have occurred along a wide trench segment? Do I misunderstand this? Please clarify.

Again, rewording of this section has hopefully made this a clearer to the reader. We also raise the point later in the paper that the evidence that FAB formation lasts for ~2 Myr arc-wide could still be consistent with the idea that the transform fault "unzipped" so long as sinking occurs rapidly enough within in a single 2D segment (<<2Myrs), which it does in our model.

Lines 115-117: This sentence, as formulated now, is not true: Subduction initiation has been modelled already previously even without any imposed boundary conditions (e.g., Cramer and Tackley 2016 or Davaille et al. 2017).

The reviewer is correct. This sentence was not specific enough and has been altered.

Line 142: Use Myr instead of Myrs.

All instances of Myrs and KYrs/Kyrs, in both the text and figures, have been changed to Myr and Kyr respectively.

Line 158: Specify what actually changes (as subduction initiation remains subduction initiation), e.g., the SI dynamics, or the SI timing, etc.

We have extended this section to include more description of the horizontally pushed model. We have also added figure S2 to figure 2 for a clearer comparison.

Line 158: The compressive force is now an "external force"? I think that is correct, but the authors claimed elsewhere that these forces are internal instead (see main comment).

We hope that our edits in response to the previous points regarding internal and external forces make the distinction between the internally generated push of the two plates against each other and the externally applied push force clearer.

Line 164: Duarte's model for SI does not make a good example here, as he argued, I think with delamination being the main cause for subduction to initiate off coast Portugal and not compression. Where would such a compression come from anyway there?

The reviewer makes a good point and Portugal has been removed as an example of subduction initiation here.

Line 166: This sentence appears problematic, since the authors don't make it clear what external and

internal forces are exactly. In addition, making such a statement based on what, two models, seems a bit overly confident.

We hope that the clarification of what we mean by internal and external forces and the clarification that we are modelling a cross section through a rapidly “unzipping” transform fault make this conclusion less problematic to the reviewer. We believe that this conclusion is a valid one to make from our results, even from two models (although we do make it clearer now that a suite was run). The fact that increasing the external push force on the system, to the point where it dominates, shuts off the formation of FAB, is robust.

Line 170: The term “large scale” - even when given with a length scale - needs to be specified somewhere. Why not just saying something like “...such events are inducing a new subduction trench simultaneously along a xxx km wide segment”. I think that would make it much clearer.

“large scale” has been done away with and replaced with the more specific “whole-plate scale”.

Line 174: I don’t think what the authors showed here in their models was actually “self-driven”, as they imposed external compression via side boundary conditions.

We hope that our clarification of what the different forces applied to our model represent, as well as the clearer distinction between forces “internal” to our 2D model and forces “internal” to the IBM as a whole, mean that the use of the term “self-driven” makes more sense.

Line 192 and other instances: Use “Myr” and not “Myrs”

All instances of Myrs and KYrs/Kyrs, in both the text and figures, have been changed to Myr and Kyr respectively.

Line 210-212: So, this is actually a model for subduction progression, not initiation? I think that is fine and useful, but probably should be clarified throughout the main part of the manuscript.

We hope that the explanation of the origin of the applied pull force earlier in the manuscript is sufficient clarification.

Figure 2:

The abbreviation for „years“, or „annum“ is somewhat undefined in the Geosciences. Yet, „Yrs“ (i.e., a plural form) is certainly not a dimension. Consider using one of the other commonly used dimensions, e.g., „y“ or „yr“ (i.e., „ky“ or „kyr“) for time intervals, as the authors actually already do elsewhere (e.g., line 103).

All instances of Myrs and KYrs/Kyrs, in both the text and figures, have been changed to Myr and Kyr respectively.

(also Figure S2) From the figure it appears that the surface temperature is 400 degree C. Is that actually the case?

The surface is at 0°C, all material below 400°C just has the same colour. The scale bar labels have been changed such that “400°C” is now “<400°C” to clarify this.

Table S1: Use „Myr“ here too.

All instances of Myrs and KYrs/Kyrs, in both the text and figures, have been changed to Myr and Kyr respectively.

Figure S2: What are the green (panel c) and black (panel d) solid lines representing?

Figure S2 has now been incorporated into Figure 2, the caption of which contains the necessary explanation.

Fabio Crameri
07.08.2019

Reviewer #2 (Remarks to the Author):

This is an interesting and timely paper discussing dynamics of subduction initiation and associated magmatism on the basis of 2D numerical models. Modeling results are compared to observations from IBM subduction system. It is demonstrated that only spontaneous (i.e., weakly/moderately forced) subduction initiation across a transform fault is capable to reproduce the IBM magmatic record (FAB=>boninites) whereas induced (i.e., strongly forced) subduction initiation scenario lacks the early decompression melting phase that is necessary to produce FAB. The paper is well written and is of broad interest but needs some reorganization and improvements:

1. Fig. S2 should be moved to the main text since it is as important as Fig. 2. Both “spontaneous” and “induced” reference models should be described in the main text to illustrate the major conclusion.

Figure S2 has been added to figure 2 and the discussion of the horizontally forced model (previously “induced”) has been extended in the main text.

2. Sensitivity of the modeling results to variations in model parameters (initial model geometry and thermal structure, rheology, faults width etc.) should be described. All models run for this paper needs to be described in the supplement (a Table with models parameters and results). Otherwise the paper creates an impression that only two shown models were run that is definitely insufficient in light of previous systematic studies (e.g., Hall et al., 2003; Gerya et al., 2008; Dymkova and Gerya, 2013, Nikolaeva et al., 2008; Zhu et al., 2009, 2011 etc.) that showed strong sensitivity of results to various model parameters.

We now fully describe the full model suite that we ran with different magnitudes for the external push force and additional slab pull force in the supplementary material as the reviewer suggests and we make it clearer in the main text that this is what we did.

With regards to varying other model parameters, please see our response to reviewer 1 who also raises this point.

3. Previous numerical modeling literature overview needs some improvements since some of the claims made for previous studies do not seem entirely correct.

Specific comments on the paper are given below.

We describe how we address the specific inadequacies of our literature review, highlighted by the reviewer, below.

Specific comments

Lines 77-80. “While this conceptual model has been partially tested dynamically (Dymkova and Gerya, 2013; Leng and Gurnis, 2011; Leng et al., 2012; Zhou et al., 2018), none reproduce the rapid time scale that the recent IBM data require, nor have they been shown to produce the very distinct magmatic sequence that Exp. 352 recovered.” It would be good to give more details concerning the

timescales. Transform collapse time scales can be very rapid (few Myr or less, e.g. Gerya et al., 2008; Gerya, 2019, Chapter 17 or Gerya, 2019, Chapter 21, subduction example). Magmatic record was to some degree explored by Nikolaeva et al. (2008) and Zhu et al. (2009; 2011), however for fluid fluxed melting only (without decompression melting).

The reviewer is exactly right that, in fact, some previous modelling work has successfully reproduced the rapid timescale of subduction initiation required at the IBM. We have adjusted this paragraph such that this is now clear and instead emphasise that we are also now able to exactly reproduce the rock record both temporally and spatially.

Lines 88-90. "We chose to use stress boundary conditions rather than a prescribed velocity as behaviour in such a large system is more likely to be governed by internal rather than external forces." Strictly saying, in case of imposed external stresses, subduction initiation regime is not purely spontaneous, since initiation is driven both by local buoyancy contrast and external forcing.

The reviewer makes a good point here. We have already dropped the use of the words "spontaneous" and "induced" in response to the comments of reviewer 1 and we hope that this removes this inconsistency satisfactorily.

Figure 2: "Temperature profiles taken at 150, 590, 790 and 920 kYrs from the start of the model run. " Rapid slab dynamics shown on this figure seems similar to these in previous studies (e.g. Gerya et al., 2008; Gerya, 2019, Chapter 17 or Gerya, 2019, Chapter 21, subduction example; Nikolaeva et al., 2008; Zhu et al., 2009; 2011; Zhou et al., 2018).

See response to a previous similar comment above.

Lines 141-142 "Crucially, the time between on the onset of FAB magmatism and the onset of boninitic magmatism is approximately 0.5 Myrs (Fig. 2a,c), in perfect agreement with drill core data (Reagan et al., 2019)." How does this time depend on model parameters (rheology, thermal/compositional structure etc.). The dynamics of the initiation is strongly sensitive to these (e.g., Hall et al., 2003; Gerya et al., 2008; Dymkova and Gerya, 2013)

We recognise that the dynamics of this system will be sensitive to a large number of model inputs. For the purposes of this study we set out to demonstrate that the stern-type model of subduction initiation was consistent with the magmatic record recovered at the arc for best estimate values for these inputs. The one model property that we do vary is the magnitudes of the driving forces to test the hypothesis they exert a first order control whether or not FAB is generated. In the future we hope to run a full parameter study of this system to quantify the effect of other parameters on FAB generation but do not feel it necessary to support the conclusions of this paper.

Supplement. Lines 100-103 "At the relatively shallow depths that we are interested in, the crustal solidus is the most significant pulse of water loss from the slab, after the early loss of pore fluid (van Keken et al., 2011), and we assume here that slab fluids migrate vertically upwards." What about serpentine dehydration? How fast does the fluid move? Is this motion affected by the mantle flow as

in previous models (e.g., Gerya et al., 2008; Nikolaeva et al., 2008; Zhu et al., 2009, 2011)

We treat fluids in a simple manner. We assume that fluids migrate instantaneously and vertically to the surface regardless of mantle flow. We deem this an appropriate method to gain a reliable time progression of magmatism as this depends on the nature of decompression melting in the mantle wedge, the timing of when the slab begins to dehydrate and the timing of the onset of sediment melting. The fact that we also match well the spatial distribution of the magmatic units by using our approximation for fluid migration perhaps indicates that this simple approach is reflecting what happens in reality. However it would be interesting to test the effect of using different fluid migration techniques (such as the method used in Dymkova and Gerya 2013) on boninite generation in future work.

Reviewer #3 (Remarks to the Author):

Review of the Manuscript #NCOMMS-19-25654-T, entitled “Dynamics of catastrophic plate boundary formation in the western Pacific”, by Maunder et al.

In this manuscript, the authors investigate the problem of subduction initiation using advanced numerical models coupled with natural observations from the long Izu-Bonin-Mariana subduction system. The forearc of this subduction zone has been recently drilled by the IODP and has been a key site for our understanding of how subduction zones initiate. Subduction initiation is a hot topic in Earth sciences. We still do not understand well how subduction zones initiate, and since subduction zones are the main driver of plate tectonics and mantle convection this understanding is crucial for our understanding of the Earth system.

I understand that the authors present arguments for spontaneous subduction initiation along a pre-existent weak (fracture) zone and suggest that such mode of subduction initiation is required to explain the observations. Alternative models have proposed that most of the subduction systems are forced by external forcing. Their proposal seems to be based on robust assumptions, observations and modelling results. I believe this will be a strong contribution to the discussion and will cause some controversy (in a positive sense), which I think is essential to keep the discussion on this topic going and further improve our knowledge. Therefore, the impact on the community is expected to be high.

The manuscript is very well written and the science is of top quality. I really enjoyed reading this paper and I find it a timely contribution. I, therefore, recommend it for publication in Nature Communications.

We thank the reviewer for their positive comments.

Minor comments:

- Please consider including the term “subduction initiation” in the title instead of “plate boundary”. I believe this may increase the impact of the paper as people will immediately know that it deals with subduction initiation.

We have adjusted the title to: Dynamics and Magmatism of Rapid Subduction Initiation in the Western Pacific.

- Line 141: typo “on”

This word has been removed.

- Line 142: avoid terms such as “perfect”. They are too subjective.

This word has been removed.

Joao Duarte

REVIEWERS' COMMENTS:

Reviewer #1 (Remarks to the Author):

Except one (see below), I think the authors did take all of my other comments seriously and have adjusted and clarified the manuscript accordingly to their best knowledge. In its present form, I think the manuscript is therefore a valuable contribution that brings us one step closer to understanding subduction zone initiation on the recent Earth. Nice paper!

- In contrast to what the authors claim in their review response, the custom dimensions for years „Yrs“ (i.e., a plural form of a dimension) is still used throughout the text and figures. Such quibbling over semantics may seem petty from my side, but then again, if we scientists keep making up definition-free dimensions and terms the majority does not agree upon, then we might indeed end up being able to talk to ourselves and ourselves only:

<https://blogs.egu.eu/divisions/gd/2018/05/09/to-serve-geoscientists/>

Fabio Crameri

10.12.2019

Reviewer #2 (Remarks to the Author):

The Authors did a good job with revisions and the paper is now suitable for publication.

Taras Gerya, Zurich, 19.12.2019